# Pharmacological Inhibition of IKK to Tackle Latency and Hyperinflammation in Chronic HIV-1 Infection

**DOI:** 10.3390/ijms232315000

**Published:** 2022-11-30

**Authors:** Ifeanyi Jude Ezeonwumelu, Edurne Garcia-Vidal, Eva Riveira-Muñoz, Eudald Felip, Lucía Gutiérrez-Chamorro, Ignasi Calba, Marta Massanella, Guillem Sirera, Bonaventura Clotet, Ester Ballana, Roger Badia

**Affiliations:** 1IrsiCaixa AIDS Research Institute, Hospital Universitari Germans Trias i Pujol, Can Ruti Campus, 08916 Badalona, Spain; 2Health Research Institute Germans Trias i Pujol (IGTP), Hospital Universitari Germans Trias i Pujol, Universitat Autònoma de Barcelona, 08916 Badalona, Spain; 3Medical Oncology Department, Catalan Institute of Oncology, Badalona-Applied Research Group in Oncology (B-ARGO), 08916 Badalona, Spain; 4University of Vic-Central University of Catalonia (UVic-UCC), 08500 Vic, Spain; 5Centro de Investigación Biomédica en Red de Enfermedades Infecciosas (CIBERINFEC), 28029 Madrid, Spain; 6Fight Infections Foundation, Hospital Universitari Germans Trias i Pujol, 08916 Badalona, Spain

**Keywords:** HIV-1, latency reversing agents, IKK, TANK binding kinase-1

## Abstract

HIV latent infection may be associated with disrupted viral RNA sensing, interferon (IFN) signaling, and/or IFN stimulating genes (ISG) activation. Here, we evaluated the use of compounds selectively targeting at the inhibitor of nuclear factor-κB (IκB) kinase (IKK) complex subunits and related kinases (TBK1) as a novel pathway to reverse HIV-1 latency in latently infected non-clonal lymphoid and myeloid cell in vitro models. IKK inhibitors (IKKis) triggered up to a 1.8-fold increase in HIV reactivation in both, myeloid and lymphoid cell models. The best-in-class IKKis, targeting TBK-1 (MRT67307) and IKKβ (TCPA-1) respectively, were also able to significantly induce viral reactivation in CD4+ T cells from people living with HIV (PLWH) ex vivo. More importantly, although none of the compounds tested showed antiviral activity, the combination of the distinct IKKis with ART did not affect the latency reactivation nor blockade of HIV infection by ART. Finally, as expected, IKKis did not upregulate cell activation markers in primary lymphocytes and innate immune signaling was blocked, resulting in downregulation of inflammatory cytokines. Overall, our results support a dual role of IKKis as immune modulators being able to tackle the HIV latent reservoir in lymphoid and myeloid cellular models and putatively control the hyperinflammatory responses in chronic HIV-1 infection.

## 1. Introduction

The human immunodeficiency virus (HIV) reservoir is established early after HIV infection through a complex and multifactorial process [1]. Although several latency reversing strategies have been proposed, none have provided clinical benefits so far. New strategies targeting the integrated HIV provirus, either directed to latency reversal or to the permanent silencing of HIV are, therefore, one of the major research priorities to HIV cure [2,3]. The HIV reservoir comprises different lymphocyte T populations [4] but also other less studied cell types from the myeloid lineage. Understanding the contribution of myeloid cells to HIV persistence is also necessary to achieve a complete HIV remission [5,6].

Innate immunity plays an important role in the early control of HIV infection, and cumulative evidence suggests a link between functional innate immunity and interferon (IFN)-mediated responses and the formation, composition, and long-term maintenance of the HIV reservoir [7]. HIV-1 transcription is a multifactorial process that requires the recruitment to the nucleus of nuclear factor-κB (NF-κB) transcription factor, among others, to initiate viral transcription [8,9]. In this process, NF-κB phosphorylation is mediated by the inhibitor of the nuclear factor-κB (IκB) kinase (IKK) complex, which is tightly regulated through the specific binding of distinct regulatory factors. The IKK family is a master regulator of many biological processes, including cell growth, metabolism, apoptosis, or cell cycle [10,11,12,13]. The NF-κB pathway and its regulatory counterparts play a complex role during the replication of primate lentiviruses. NF-κB is essential for the induction of efficient proviral gene expression, but also contributes to the innate immune response and induces the expression of numerous cellular antiviral genes. Recent data suggest that primate lentiviruses cope with this challenge by boosting NF-κB activity early during the replication cycle to initiate Tat-driven viral transcription and, by suppressing it at later stages to minimize antiviral gene expression [14]. IKK-related kinases Tank binding kinase 1 (TBK1) and IκB kinase-ε (IKKε) are confluent downstream regulators of pattern recognition receptors (PRRs) signaling pathways involved in the innate sensing of foreign nucleic acids. Therefore, NF-κB-IKK interplay becomes a key target for viruses (i.e., HIV), which have evolved mechanisms to modulate the NF-κB–IkB-IKK crosstalk to favor their replication cycle [15]. Currently, several inhibitors of IKK complex and the IKK-related kinases are under development for the treatment of cancer and inflammatory diseases [14,16,17,18]. Taking advantage of our previous experience in the screening of compound libraries [19,20,21], we explored the role of IKK inhibitors (IKKis) as potential latency reversing agents (LRAs) in the context of HIV chronic infections. 

## 2. Results

### 2.1. IKKis Induce HIV Reactivation in Distinct Models of HIV Latency In Vitro

To assess the latency reactivation capacity of IKKis, we evaluated their role in lymphoid and myeloid models of HIV latency (summarized in Appendix A). The LRA activity of selected IKKis was evaluated according to their highest subtoxic concentration for each specific compound of interest. Since toxicity is a major concern for the use of histone deacetylase inhibitors (HDACis), we used them at subtoxic concentrations as previously described [22]. Compared to HDACis, IKKis showed low cytotoxicity at micromolar level in the J-HIG and HL-HIG latency models (Figure 1A,B). Six of the tested inhibitors were able to induce viral reactivation in both lymphoid and myeloid cell models (Figure 1A,B), without significantly compromising cell viability with the exception of INH2 at 20 µM in the myeloid model. HIV-1 latency reversing activity of IKKis at the micromolar level was comparable to subtoxic concentrations of HDACi of panobinostat (PNB) and vorinostat (VOR) in both models. These results were confirmed in latently infected ACH-2 cells by measuring HIV p24 Ag in the supernatant (Appendix A). 

The selective TBK/IKKεi MRT67307 is an improved version of BX-795 with less off-targets described effects [12] and TPCA-1 is a selective IKKβi, both representing a 1.6- and 1.8-fold viral reactivation increase at the highest concentrations tested (respectively *p* < 0.01, compared to the non-treated control). Therefore, they were selected as representative IKKis for further characterization of their LRA potential.

### 2.2. TBK1/IKKε and IKKβ Inhibitors Induce HIV Reactivation Ex Vivo in CD4+ T Cells from HIV+ Individuals

Viral reactivation capacity of the representative IKK inhibitors was evaluated ex vivo in CD4+ T cells from HIV+ individuals (n = 3, Appendix A). Both IKKis, MRT67307 and TPCA-1, induced HIV-1 reactivation in all participant samples similar to the positive control phorbol-12-myristate-13-acetate (PMA)/ionomycin. Despite the variability in HIV-RNA abundance between samples, HIV-1 reactivation was observed in all patients for MRT67307 (2.1–2.2-fold induction) and TPCA1 (1.3–3.5-fold change) compared to untreated cells (Figure 1C). 

### 2.3. Antiretroviral Drugs Do Not Interfere with HIV Reactivation Activity Induced by IKKis

In clinical settings, LRAs have to be combined with ART to avoid new rounds of infection. IKKis showed no antiviral activity in in vitro models (Appendix A). To evaluate the possibility of administering IKKis with ART, we evaluated the potential interaction between MRT67307, TPCA-1, and the RT inhibitor efavirenz (EFV) and the integrase inhibitor raltegravir (RAL) at concentrations where EFV and RAL harbor anti-HIV activity [23]. Neither EFV nor RAL modified the IKKis LRA activity in J-HIG and HL-HIG (Figure 1D). Accordingly, IKKis did not modify the anti-HIV activity of ART (Appendix A). Therefore, studied IKKis did not show any synergistic or antagonistic effect when combined with antiretroviral drug and their reactivation capacity was neither modified, indicating the feasibility of combinations with the current antiretroviral regimens.

### 2.4. IKKis Do Not Modify Cell Activation in Ex Vivo Treated Primary CD4+ T Cells 

The effect of IKKis on cell activation and innate immune signaling was also evaluated. Interestingly, treatment with MRT67307 or TPCA-1 blocked cytokine expression of *IL1-β*, *IL8*, *CXCL10* and *TNFα* (30% to 80% reduction depending on each cytokine; *p* < 0.05 and *p* < 0.001 respectively), in contrast to the PMA control (Figure 2A). Similarly, treatment of peripheral blood mononuclear cells (PBMCs) with MRT67307 or TPCA-1 did not trigger pro-inflammatory cytokine production (IL1-β, IL-6, TNFα), contrary to PMAi or LPS, which significantly induced the expression of IL-1β (LPS *p* < 0.001), IL-6 (LPS *p* < 0.01) and TNFα (LPS *p* < 0.05; PMAi *p* < 0.001) compared to the ND control at subtoxic concentrations (Figure 2B). Evaluation of the innate immune signaling activation confirmed the reported effects on targeted pathways, characterized by decreased phosphorylation of STAT1 together with decreased expression of melanoma differentiation-associated gene 5 (MDA5), although results for this intracellular RNA sensor belonging to the Retinoic acid-inducible gene I-like receptors (RLRs) family were statistically non-significant (Figure 2C). Moreover, IKKis did not promote the expression of cell surface activation markers CD25, CD69, and HLA-DR ex vivo in CD4+ T cells from HIV-negative donors (Figure 2D,E). Indeed, TPCA-1 treatment reduced the expression of CD25 (72% decrease, *p* < 0.001) and HLA-DR (30% decrease, *p* < 0.05). 

## 3. Discussion

The IKK-related kinases TBK1 and IKKε play prominent roles in mediating IFN production upon pathogen recognition, leading to activation of relevant innate immune mediators, including transcription factors, IFN stimulating genes (ISGs), and induction of antiviral IFN-JAK-STAT signaling pathway, being, therefore, putative therapeutic targets for modulating innate immunity. 

Innate immunity in HIV-1 pathogenesis is best understood in the context of acute infection. However, recent data suggest that innate immunity can also be used to improve the efficacy of HIV-1 cure strategies [24]. Here, we characterize the latency reactivation capacity of commercially available IKKis. In our study, we postulate that regulation of IKK-TBK inhibition may also represent a novel therapeutic intervention to revert HIV latency, which also has the potential to avoid the unwanted autoimmune- and or inflammation-related side effects associated with latency-reversing strategies. We demonstrate that IKKis have the capacity to reverse HIV-1 latency in lymphoid and myeloid cellular models. Accordingly, NF-κB modulation, through siRNA silencing of IkBα and NF-κB binding to the kB sites of the HIV-LTR, was previously described to activate HIV in latently infected monocytic U1 and lymphoid J-Lat 10.6 [10,25]. These results were confirmed in ex vivo CD4+ T cells from HIV+ individuals by ultra-sensitive semi-nested qPCR that revealed high IKKis-induced viral transcription comparable to PMA and ionomycin reactivation, as previously reported [26]. These results indicate the importance of the inhibition of the TBK-IKK, as key regulators of NF-κB and interferon regulatory factors 3 and 7 (IRF-3/IRF-7) [14] in the context of HIV latency. Our data are in clear contrast to the well-described role of NF-κB as a transcription factor affecting HIV-1 transcription. However, several evidences from the literature support the idea that the shock phase of the shock-and-kill approach to reverse HIV-1 latency may be achieved in the absence of NF-κB [27]. Moreover, other transcription factors have also been demonstrated to participate in HIV reactivation, resulting in a far more complicated process than the oversimplified LTR-driven transcription by NF-κB [28,29]. Finally, compensatory IKK regulatory mechanisms cannot be excluded [12]. An in-depth description of the underlying mechanisms controlling IKKis capacity as LRA is warranted in future studies.

Novel LRAs must induce the transcription of latent HIV without triggering global cell immune activation or chronic inflammation [30]. In one of the earliest studies investigating the potency of TLR agonists as LRAs, Novis et al. reported that the TLR1/2 agonist Pam3CSK4 reactivates HIV-1 transcription via NF-κB and NFAT-dependent pathways, but without induction of IFNs or T cell activation [31]. Although Pam3CSK4 showed less potency compared to the HDACi panobinostat, this finding lends credence to the possibility of harnessing immune modulation as a strategy to reverse HIV-1 latency and circumvent immune cell activation. The modulation of NF-κB by cytokines/chemokines or PKC agonists has been discouraged due to drug tolerability concerns [25,32,33]. Alternatively, the LRA activity of IKKis had minimal impact on cell activation markers in primary CD4+ T cells. Importantly, IKKi-induced HIV reactivation was accompanied with a partial block of the innate immune signaling and decreased induction of proinflammatory cytokines such as IL8, IL1β, and the chemokine CXCL10. 

In conclusion, IKKis targeting TBK1 and IKKε present significant HIV-1 reactivation capacity in lymphoid and myeloid cellular models. The low toxicity in the absence of cell activation suggests their use as alternative LRAs targeting both lymphoid and myeloid compartments avoiding hyperinflammatory responses in chronic HIV-1 infection.

## 4. Materials and Methods

### 4.1. Cells 

PBMCs were obtained from the blood of healthy donors using a Ficoll-Paque density gradient centrifugation and CD4+ T lymphocytes were purified using negative selection antibody cocktails (StemCell Technologies, Vancouver, BC, Canada) as described previously [34]. Buffy coats were purchased from the Catalan Banc de Sang i Teixits. The buffy coats received were totally anonymous and untraceable and the only information given was whether or not they have been tested for disease. PBMCs or CD4+ T lymphocytes were kept in complete RPMI 1640 medium supplemented with 10% heat-inactivated fetal bovine serum (FBS; Gibco, Billings, MT, USA), 100 U/mL penicillin, and 100 µg/mL streptomycin with IL-2 (16 U/mL) and then treated with the corresponding compounds for 48–72 h.

The human cell lines Jurkat, HL-60 and ACH2 (AIDS Reagent Program, National Institutes of Health, NIH, Bethesda, MD, USA) were cultured in RPMI 1640 medium (ThermoFischer, Madrid, Spain) supplemented with 10% heat inactivated FBS, 100 U/mL penicillin and 100 µg/mL streptomycin.

HIV+ individuals: All participants in the study provided informed consent and samples were processed as mentioned above. Subject samples were included if the individuals were older than 18 years old, had chronic HIV-1 infection, and had previously been on highly active ART for >1 year. HIV-RNA levels were >110,000 viral load (VL) pre-ART copies/mL, and <40 copies/mL during for at least 2 years at study entry. The immunological and virological characteristics from all participants are found in Appendix A.

### 4.2. Compounds 

MRT67306 and BX795 were purchased from Invivogen (Ibian Technologies, Zaragoza, Spain). TCPA-1 (S2824), TBK1/IKK-IN-1 (S8922), TBK1/IKK-IN-2 (S0425), SC514 (S4907), and GSK8612 (S8872) were purchased from Selleckchem (Munich, Germany). Vorinostat was purchased from Prochifar srl (Lombardy, Italy), and panobinostat was purchased from LC Laboratories. Antiretroviral compounds raltegravir (RAL) and efavirenz (EFV) as well as lipopolisaccaride (LPS) and PMA were obtained from Sigma-Merck, Burlington, MA, USA. All compounds were reconstituted in dimethyl sulfoxide (DMSO) and stored at −20 °C until use.

### 4.3. Virus 

Envelope-deficient HIV-1 NL4-3 clone encoding IRES-GFP (NL4-3-GFP) was pseudotyped with VSV-G by cotransfection of HEK293T cells using polyethylenimine (Polysciences, Warrington, PA, USA) as previously described [34,35]. Three days after transfection, supernatants were harvested, filtered, and stored at −80 °C. Viral stocks were concentrated using Lenti-X concentrator (Clontech, Mountain View, CA, USA). Viruses were titrated by infection of Jurkat or HL-60 cells followed by GFP quantification by flow cytometry.

### 4.4. Generation of Latently Infected Cells 

Latently infected Jurkat (J-HIG) and HL-60 (HL-HIG) cells were generated by following a modification of the protocol described by Li et al. [36,37]. Briefly, cells were generated after acute infection of CD4+ Jurkat or HL-60 cells with VSV-pseudotyped HIV-1 NL4-3-GFP and were maintained in culture for 10 days to allow the attrition of productively infected cells. 

### 4.5. HIV Reactivation In Vitro in Latently Infected Cells 

HIV reactivation was measured as described previously [37]. Briefly, J-HIG or HL-HIG cells were incubated for 24 h with different concentrations of IKKIs. HDACi PNB and VOR were used as controls for HIV-1 reactivation. Reactivation of HIV was monitored as the percentage of living GFP+ cells by flow cytometry.

Similarly, ACH-2 cells were cultured in the presence or absence of IKKis, PNB, or VOR for 48 h at 37 °C and 5% CO_2_. HIV reactivation was measured by the production of HIV CA p24 antigen in the supernatant using Genscreen HIV-1 Ag ELISA (BioRad, Hercules, CA, USA) according to manufacturer’s instructions.

### 4.6. Measurement of HIV Production in CD4+ T Cells from HIV+ Individuals

To assess HIV-1 reactivation, CD4+ T cells from HIV+ individuals were pre-incubated with 10 µM of pan-caspase inhibitor Q-VP-Oph (Merck, Ref. SML0063) for 2 h at 37 °C and 5% CO_2_. Then, 3.5–5 × 10^5^ cells were plated/treatment condition with the LRAs compounds/controls. The medium was supplemented with recombinant IL-2 (rIL-2) (30 IU/mL; Roche) and antiretroviral drugs RAL (2.2 µM) and EFV (0.3 µM) to avoid multiple cycles of HIV reinfection. Sample processing and HIV production measurement by ultrasensitive semi-nested RT qPCR were performed as previously described [21,38]. 

### 4.7. Quantitative RT-Polymerase Chain Reaction (qRT-PCR)

Total RNA was extracted using NucleoSpin RNA II kit (Ref. 740955, Macherey-Nagel, Düren, Germany) as recommended by the manufacturer. Reverse transcription was performed using the PrimeScript™ RT-PCR Kit (RR036A, Takara, Kusatsu, Japan) following the manufacturer’s instructions. mRNA levels of all genes were measured by two-step quantitative RT-PCR and normalized to GAPDH mRNA expression using the DDCt method. The following primers and DNA probes were obtained from TaqMan Gene expression assays (Thermofisher Scientific, Waltham, MA, USA): GAPDH (Hs00266705_g1), CXCL10 (Hs00171042_m1), IL-1β (Hs01555410_m1), IL8 (Hs00174103_m1), and TNFα (Hs00174128_m1). 

### 4.8. Western Blot 

The treated cells were rinsed, lysed, subjected to SDS-PAGE, and transferred to a polyvinylidene difluoride (PVDF) membrane, as previously described [37]. The following antibodies were used for immunoblotting: anti-rabbit and anti-mouse horseradish peroxidase-conjugated secondary antibodies (1:5000; Pierce, Dallas, TX, USA); anti-GAPDH (1:2500; ab9485; Abcam, Cambridge, UK); anti-phosphoSTAT1 (1:1000; 9167, Cell Signalling, Danvers, MA, USA), anti-MDA5 (1:500; 5321, Cell Signalling). Blots were immersed in chemiluminescent substrate (SuperSignal West Pico Plus or Femto, Thermo Fisher Scientific), and the signal was visualized using ChemiDoc MP imaging system (BIORAD, Hercules, CA, USA).

### 4.9. Flow Cytometry

Cells were labeled with the following anti-human antibodies: CD3 FITC (345763), HLA-DR PE-Cy™7 (335830) purchased from BD and CD4 BV786 (317442), CD25 APC (302610), and CD69 BV650 (310934), all obtained from Biolegend (Palex, Spain). Cells were incubated for 40 min at room temperature in the dark. Then, cells were washed with phosphate buffered saline (PBS) and fixed with 1% formaldehyde (FA). 

For intracellular assessment of cytokine production, 1 × 10^5^ PBMCs/well were seeded in a polypropylene round-bottom 96-well plate in the presence of the compounds. Two hours after, protein transport inhibitors GolgiPlug (5 µL/mL) and GolgiSTOP (0.66 µL/mL) (Becton Dickinson, Franklin Lakes, NJ, USA) were added and incubated overnight. Cells were then washed with PBS and incubated at 4 °C with BD Cytofix/Cytoperm (Becton Dickinson, USA) for 45 min. Afterwards, cells were washed twice with BD Perm/Wash (Becton Dickinson, USA) diluted 1:10 in water, incubated at 4 °C with the following antibodies (all from BioLegend, Palex, Spain): PE anti-human IL-6 (501107), FITC anti-human IL-1β (511705), Brilliant Violet 650 anti-human TNF-α (502938) (1:50 in diluted Perm/Wash) for 1 h, washed twice again in diluted Perm/Wash and resuspended in formaldehyde 1% in PBS. Cells were analyzed by flow cytometry (BD FACSCelesta™ cell analyzer, BD, Franklin Lakes, NJ, USA) on the same day.

Flow cytometry was performed in a FACS LSRII flow cytometer (BD Biosciences, Franklin Lakes, NJ, USA). The data were analyzed using the FlowJo™ Software Version 10.6.1. (Becton Dickinson, 2019). Analysis of the global immune profile of PBMCs was done using OMIQ data analysis software (www.omiq.ai, last access on 24 September 2022). Cells were gated on CD3+CD4+ singlets and individual files were concatenated and clustered as a whole in each treatment condition using the optimized t-distributed stochastic neighbor embedding (opt-SNE) algorithm for dimensional reduction and visualization.

### 4.10. Evaluation of Cytotoxicity

Cells were treated at the indicated doses of the test compounds between 24 to 72 h (according to each specific experiment) and then stained for 30 min with LIVE/DEADTM Fixable Green Dead Cell Stain Kit (Invitrogen, Waltham, MA, USA, Thermofisher Scientific) in PBS according to the manufacturer’s instructions. Alternatively, viable cells were identified according to forward and side laser light scatter flow cytometry analysis, as described [39]. 

### 4.11. Anti-HIV Assays

Antiviral activity of TBK1 and IKKβ inhibitors was assessed on Jurkat and HL-60 cells infected with pseudotyped HIV-1, expressing a GFP reporter gene (VSV-HIG). Untreated condition and antiretroviral compounds RAL and EFV were included as controls. Antiviral activity was measured 48 h after infection as measured as the expression of GFP by flow cytometry. Determinations were performed in triplicates and data calculated from three independent experiments.

### 4.12. Statistical Methods

Data were analyzed with the PRISM statistical package. If not stated otherwise, all data were normally distributed and expressed as mean ± SD. *p*-values were calculated using an unpaired, two-tailed, *t*-student test.

## Figures and Tables

**Figure 1 ijms-23-15000-f001:**
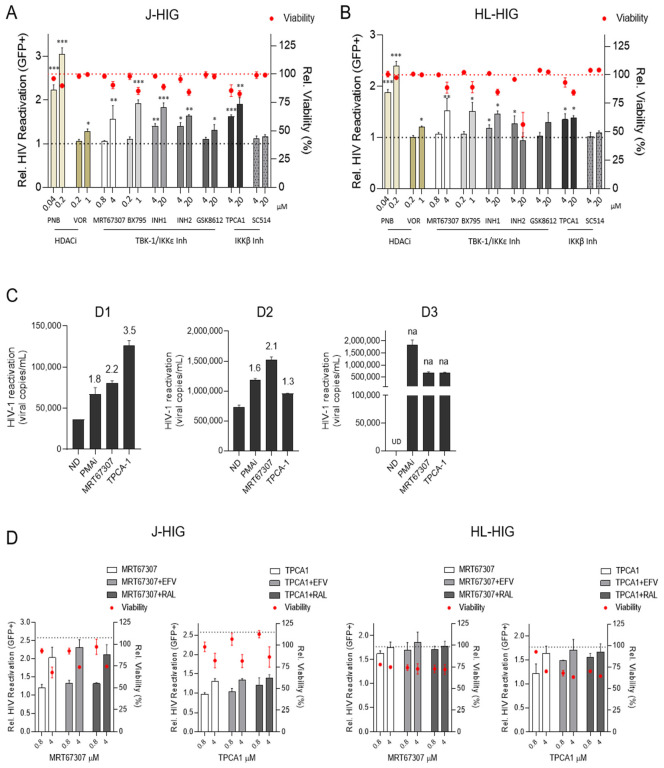
IKK Inhibitors induce HIV reactivation in in vitro and ex vivo models of latency. (**A**) HIV reactivation induced by IKKis in latently infected lymphoid Jurkat (J-HIG) and (**B**) myeloid HL-60 (HL-HIG) cells. The activity of IKKis MRT67307, TPCA-1, BX795, Inh 1, Inh2, GSK8612 and SC514 was determined by the quantification of GFP+ cells (%) after culturing J-HIG or HL-HIG cells with IKKis for 24 h. HDCAi panobinostat (PNB) and vorinostat (VOR) were used as controls for HIV-1 reactivation. Basal reactivation (grey-dashed line) in J-HIG and HL-HIG was established according to the no-drug condition (ND). Toxicity of IKKis (red-dashed line) in J-HIG and HL-HIG cells was measured as percentage of viable cells (LIVE/DEAD™ staining) by flow cytometry related to the ND condition. (**C**) Ex vivo viral reactivation capacity of IKKis in CD4+ T cells from HIV+ individuals. HIV-RNA copies were determined in the cell culture supernatant from HIV+ CD4+ T cells incubated for 72 h with 4 µM of MRT67307 or TPCA-1. PMA/ionomycin (50 ng/mL/1 µM, PMAi) was used as the positive control of HIV-1 reactivation. Fold-change (FC) relative to ND are indicated above each bar for the respective treatment conditions. FC values, if not available (na) because of undetermined RT-qPCR ct values (UD), are highlighted in the chart. (**D**) HIV-1 reactivation capacity of IKKis in the presence of antiretroviral drugs. Reactivation ability of MRT67307 (4 µM) and TPCA-1 (4 µM) is not affected by the ART (efavirenz, EFV at 0.3 µM or raltegravir, RAL at 2.2 µM), neither in latently infected lymphoid J-HIG (left panel) nor myeloid HL-HIG cells (right panel). Viral reactivation was measured 24 h after incubation with the corresponding conditions. The grey-dashed line indicates reactivation in the PMAi control. Red dots indicate relative cell viability. Mean ± SD of three independent experiments is shown. * *p* < 0.05; ** *p* < 0.01; *** *p* < 0.001.

**Figure 2 ijms-23-15000-f002:**
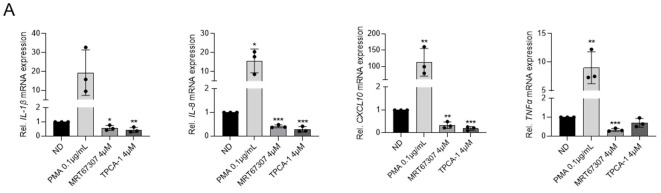
Treatment with IKKi blocks production of inflammatory signals and T-cell activation. (**A**) Gene expression of proinflammatory cytokines IL-1β, IL-8, CXCL10, and TNFα in HL-60 cells after 16 h of treatment, as measured by RT-qPCR. PMA (0.1 µg/mL) was used as positive control. (**B**) Expression of proinflammatory cytokines IL-1β, IL-6, and TNFα as measured by intracellular staining by flow cytometry after 24 h of treatment in CD4+ T cells from HIV-negative donors. Viability of CD4+ T cells treated with IKKis as measured by flow cytometry. Black dots represent individual data from independent experiments. Mean ± SD of three independent experiments is shown. (**C**) Evaluation of innate immune activation at protein level. Representative western blot (out of three) showing the inhibition of distinct ISG upon treatment with IKKi (left panel). Semiquantitative analysis of ISG expression upon treatment with IKKi (right panel). (**D**) Distribution of cell surface markers in CD4+ T cells from uninfected donors treated with subtoxic concentrations of MRT67307 (4 µM) and TPCA-1 (4 µM) measured by flow cytometry. Distribution was determined by Opt-tSNE-guided manual gating analysis of the signal strength of key phenotypic defining cell activation markers CD25, CD69, and HLA-DR from eight HIV-negative donor samples. Opt-tSNE analysis was performed using 1000 iterations, a perplexity of 30, a trade-off θ of 0.5 as implemented by OMIQ data analysis software (www.omiq.ai). (**E**) Quantification of the cell activation markers compared to resting and PMAi activated CD4+ T cells. Each symbol refers to a distinct patient. Mean ± SD of n = 8 individuals is shown. * *p* < 0.05; ** *p* < 0.01; *** *p* < 0.001.

## Data Availability

The data presented in this study are available on request from the corresponding author.

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
