# Peer review of "Pharmacological Inhibition of IKK to Tackle Latency and Hyperinflammation in Chronic HIV-1 Infection"

_ijms, 2022, doi:10.3390/ijms232315000_

Round 1
Reviewer 1 Report
General Comments
References should be cited in the text by sequential numbers in square brackets. Example [1].
Please, italicize in vitro and ex vivo, except when these are in an italicized sentence.
Graphs, panels or others within the Figures must be marked with capital letters.
Authors
Line 5. Please write the comma after Massanella1,4,5, at the same level as the main text, without superscript: Massanella1,4,5,
Abstract
Line 23. IKKis should be defined. IKKis should be added in parentheses after the written-out form.
Line 23. Please, write in vitro in italics letters
Line 26. PLWH should be defined. PLWH should be added in parentheses after the written-out form.
Introduction
Line 37. References should be cited in the text by sequential numbers in square brackets. Example [1].
Line 59. Add space before reference 14.
Results
Line 72. Please, add space after the word Table.
Line 74. Please, define the acronym HDACis.
Lines 78-79. The authors state that there is no significantly compromising cell viability in the experiments of Figures 1A and 1B. However, the viability of the HL-HIG cell line treated with 20 µM of INH2 compound is low; it does not reach 60% (Fig.1B). The authors are requested to comment on this result.
Line 79. The authors state that HIV-1 reversing activity of IKKis was higher than VOR in both models. However, IKKis are tested at concentrations higher than VOR, with the exception of BX795. So that statement may not be valid. The authors should reanalyze that results and write accordingly.
Line 79. It is suggested to add the word latency before reversing, or replace the word reversing with reactivation.
Line 80. The authors state that MRT67307 and TPCA-1 showed the best reactivation profiles, however, BX795 and INH1 showed a greater ability for HIV reactivation than MRT67307 in J-HIG cell line. Particularly, the compound BX795 showed higher HIV reactivation even at lower concentrations than MRT67307. The authors should explain why they consider the MRT67307 compound have the best reactivation profile.
Line 83. In this sentence, as in Materials and Methods (line 240) the authors state that the evaluation of HIV p24 Ag is performed on the supernatant; however, in the caption of FigS1 it is indicated that the evaluation was performed intracellularly. Please, clarify.
Line 89. The authors should define PMA as it is the first time it appears in the text.
Line 90. Please rectify the error as appropriate. Line 90 is written (2.1-2.3-fold induction) while Figure 1C (D1) shows 2.2.
Line 91. Please rectify the error as appropriate. In line 91 it is written (1.3-3.6-fold change) while Figure 1C (D1) shows 3.5.
Line 115. Add space after the word or. Delete highl.
Line 116. Replace controls with control
Line 116. NA should be written in lowercase letters in the figure caption as it is written that way in Figure 1C. Otherwise write NA in capital letters in the figures.
Figure 1C. Graphs D1 and D2 show a high number of viral copies/ml which may not be expected since donors of CD4+ cells have viral load <40 copies/ml (Table S2). The authors are suggested to comment on this aspect.
Figure 1C. The meaning of the numbers shown above each bar (graphs D1 and D2) should be written in the figure caption.
Figure 1D. It is suggested to the authors to indicate in the graphs those that correspond to the J-HIG and HL-HIG cell lines.
Line 122. Replace comma with semicolon after 0.01
Line 123. Please do not italicize ex vivo in this sentence.
Figure 2. Authors are suggested to write the full name of the compound MRT6707 in the graphs of panel A, or else define the abbreviation MRT.
Line 127. Replace PMA with PMAi.
Line 128. The authors should define PBMCs as it is the first time it appears in the text.
Line 133. Figure 2B does not show NFκB results as referred in the text. Please clarify.
Figure 2C. The graph on the right should show statistical significance.
Line 142. The authors describe in the figure caption that they used PMA at 50 ng/ml, however, in the Figure 2A they state that they used PMA at 0.1 µg/ml. Please rectify as appropriate.
Line 146. Delete *p < 0.05; **p < 0.01 since it is described at the end of the figure caption.
Line 157. Add *** p < 0.001
Discussion
The authors should improve the Discussion by commenting on their results, especially those that have been indicated in the review document (line 78, 79, 80 y Figure 1C.)
Materials and Methods
Line 191. Eliminate Peripheral blood mononuclear cells. The definition must appear the first time it is used in the text on Line 128. Remove parentheses.
Line 208. Please, define the abbreviation VL.
Line 211. Delete point after the word Compounds
Line 217. Delete Phorbol 12-Myristate 13-Acetate. The definition must appear the first time it is used in the text on line 89. Remove the parentheses around PMA.
Line 227. Delete point after the word cells.
Lines 228 and 234. Substitute J-Hig for J-HIG
Line 233. Please do not italicize in vitro in this sentence.
Line 245. Please write in superscript 105
Line 249. Eliminar paréntesis
Line 265. NFκB results do not appear in the western blot, so the information corresponding to the anti-pNFκB antibody must be removed.
Line 274. Please write in superscript 105
References
Line 342. Delete comma before 2018. Please, write 2018 in bold.
Line 345. Remove semicolon before International
Line 352. Please, replace t with T
Line 355. Remove comma after the word Reports
Line 357. Remove comma after the word Immunology
Line 359. Remove 2022 19:1
Line 361. Remove comma after the word Expression
Line 363. Remove comma after the word Retrovirology
Lines 364 y 365. Add κ where appropriate (IkB, NF-κB)
Line 368. Remove comma after the word Disease
Line 395. Remove comma after the word Microbiology
Line 397. Please replace Ballana Esther and with Ballana, E.;
Line 401. Please replace Crespan Emmanuele and with Crespan, E.;
Line 415. Please, write in vitro in italics letters
Line 422. Please replace Este Jose A. and with Esté, J. A.;
Line 424. Please replace Marti Ramon and with Martí, R.;
Line 425. Please, write Cell Cycle in lowercase letters
Line 427. Please, replace v.; with V.;
Line 433. Please replace Puig Teresa and with Puig, T.;
Supplementary Material
Table S1
Line 5. Write 160 on the line below, so that there is 160 nM on the same line.
Line 8. Add space between 158 and nm
Line 9. Please, substitute SC-514 for SC514
Figure S1
Line 66. Please, write in vitro in italics
Line 77. p significant are equals for ** or ***, please rectify.
Figure S2
Line 75. Please, write in vitro in italics
The letters a and b in each graph should appear in capital letters (A and B) to match the way they are written in the figure caption.
Figure S3
Please, modify the letters of signaling the graphics according to what is described in the figure caption.

Author Response
We thank the reviewer for the exhaustive revision and constructive comments and observations that have significantly improved the manuscript. First, all the comments regarding reference format, language and punctuation marks and formatting issues have been corrected as suggested by the reviewer. Moreover, we have gone throughout the manuscript to fix all issues regarding, language and reference formatting.
For all other comments, find below a point-by-point response:
Lines 78-79. The authors state that there is no significantly compromising cell viability in the experiments of Figures 1A and 1B. However, the viability of the HL-HIG cell line treated with 20 µM of INH2 compound is low; it does not reach 60% (Fig.1B). The authors are requested to comment on this result.
A sentence describing the effect of INH2 on cell viability has been added to the text (P2; L81-82)
Line 79. The authors state that HIV-1 reversing activity of IKKis was higher than VOR in both models. However, IKKis are tested at concentrations higher than VOR, with the exception of BX795. So that statement may not be valid. The authors should reanalyze that results and write accordingly.
According to Reviewer’s comment, the statement has been reformulated to provide a precise description of the results (P2; L82-84).
Line 79. It is suggested to add the word latency before reversing, or replace the word reversing with reactivation.
Corrected, and now written as…latency reversing…
Line 80. The authors state that MRT67307 and TPCA-1 showed the best reactivation profiles, however, BX795 and INH1 showed a greater ability for HIV reactivation than MRT67307 in J-HIG cell line. Particularly, the compound BX795 showed higher HIV reactivation even at lower concentrations than MRT67307. The authors should explain why they consider the MRT67307 compound have the best reactivation profile.
We appreciate the comment of the Reviewer. We have clarified the criteria to choose MRT67307 and TPCA-1 and reformulated the text accordingly (P3; L93-97).
Line 83. In this sentence, as in Materials and Methods (line 240) the authors state that the evaluation of HIV p24 Ag is performed on the supernatant; however, in the caption of FigS1 it is indicated that the evaluation was performed intracellularly. Please, clarify.
We appreciate the comment of the reviewer. HIV Ag p24 was determined in cell culture supernatants and this has been corrected in the caption of Supplementary Figure 1 (Suppl. Material; P3; L54-56), accordingly to the text in Results and Materials and Methods sections.
Line 90. Please rectify the error as appropriate. Line 90 is written (2.1-2.3-fold induction) while Figure 1C (D1) shows 2.2. Line 91. Please rectify the error as appropriate. In line 91 it is written (1.3-3.6-fold change) while Figure 1C (D1) shows 3.5.
Corrected.
Figure 1C. Graphs D1 and D2 show a high number of viral copies/ml which may not be expected since donors of CD4+ cells have viral load <40 copies/ml (Table S2). The authors are suggested to comment on this aspect.
PLWH included in the study have been infected for years before they become undetectable. An ultrasensitive semi-nested qPCR-based method has been used to quantify viral transcription as described by Vandergeeten C et. al, Blood 2013 (PMID: 23589672). HIV viral copies from CD4+ T cells from HIV+ individuals treated with IKKis and/or PMAi controls are similar to previously described by Laird et al., J Clin Investig 2015 (PMID: 25822022).
Figure 1C. The meaning of the numbers shown above each bar (graphs D1 and D2) should be written in the figure caption.
A description of the meaning of the numbers and “na” has been included in the text of the legend in figure 1 (P4; L147-149).
Figure 1D. It is suggested to the authors to indicate in the graphs those that correspond to the J-HIG and HL-HIG cell lines.
We agree the comment. J-HIG and HL-HIG are now indicated where appropriate in the graphs in the main manuscript and in the supplementary material.
Figure 2. Authors are suggested to write the full name of the compound MRT6707 in the graphs of panel A, or else define the abbreviation MRT.
Full name of the compound MRT6707 has been included in all graphs.
Line 133. Figure 2B does not show NFκB results as referred in the text. Please clarify.
Figure 2B does not show NFkB results and the text has been modified accordingly.
Figure 2C. The graph on the right should show statistical significance.
The statistical analysis of western-blot semi-quantification is now included in both graphs in Figure 2C. Text referring to these data has been checked accordingly (P5; L171 and P5:L173-173).
Line 142. The authors describe in the figure caption that they used PMA at 50 ng/ml, however, in the Figure 2A they state that they used PMA at 0.1 µg/ml. Please rectify as appropriate.
Corrected.
Discussion
The authors should improve the Discussion by commenting on their results, especially those that have been indicated in the review document (line 78, 79, 80 y Figure 1C.)
Following the reviewer’s suggestion, the discussion section has been improved by adding relevant data on viability and reactivation capacity, including the comparison of our ex vivo reactivation data with similar reports from the literature.
Reviewer 2 Report
This brief report manuscript, titled "Pharmacological inhibition of IKK to tackle latency and hyper inflammation in chronic HIV-1 infection" by Ezeonwumelu and colleagues, shows that application of IKK inhibitors can be applied to increase virus production from latently infected cells and do so without causing excessive inflammation. While this article makes a good case for the use of IKK inhibitors it's a bit frustrating that the authors discuss mechanisms that might be at play but don't address in any firm manner what they think is going on. Indeed it's not clear why they sought to test IKK inhibitors in the first place. The point is made in at least two places that HIV-1 responds to NF-kB, however IKKs phosphorylate I-kB to mobilize or free NF-kB to enter the nucleus and activate transcription of HIV-1 among other targets. If this is the case, we expect that IKK inhibitors would interfere with the activation of latent proviruses. As is also alluded to, that NF-kB upregulates transcription of antiviral responses. Could these be involved in decreasing latency. As this work is written it seems that somehow the investigators came upon IKK inhibitors as a means to modestly enhance expression from latent viruses and they have no idea why. If NF-kB is important for HIV-1 transcription, especially in the initial absence of Tat, the must be stronger factors that counterbalance this effect to start transcription. More discussion is important here.
Minor points:
- the abbreviation IKKis should be defined
- VOR was not defined until the figure legend of Figure 1 on page 3 although it's used on page 2.
- line 86 should read "...best-performing IKK inhibitors..." for better clarity.
- line 97 "...concentrations where harbour anti-HIV activity..." is missing something (between where and harbour?).
- Figure 1 c D3 is unclear. na?
- in the Discussion section, "modulation" should be defined for clarity. Modulation, of course, can be up- or down-regulation (line 166). NF-kB is mobilized by silencing I-kB alpha. Again, this "activates HIV" (line 168). This is of course the opposite of what would be achieved by inhibiting IKKs... here also (line 170) there is mention of "modulating" TBK. Only IKK inhibitors are used and discussed, so it would bee more clear to just say inhibition.
- Line 162: "... immunoblotting: antirabbit..." should read anti-rabbit to be consistent with the other antibody designations.
- Line 272 should read "in the dark"
- line 274 the 5 in 105 should be a superscript.
Author Response
We thank the reviewer for the exhaustive revision and constructive comments and observations that have significantly improved the manuscript. Find below a point-by-point response to the concerns raised.
This brief report manuscript, titled "Pharmacological inhibition of IKK to tackle latency and hyper inflammation in chronic HIV-1 infection" by Ezeonwumelu and colleagues, shows that application of IKK inhibitors can be applied to increase virus production from latently infected cells and do so without causing excessive inflammation. While this article makes a good case for the use of IKK inhibitors it's a bit frustrating that the authors discuss mechanisms that might be at play but don't address in any firm manner what they think is going on. Indeed it's not clear why they sought to test IKK inhibitors in the first place. The point is made in at least two places that HIV-1 responds to NF-kB, however IKKs phosphorylate I-kB to mobilize or free NF-kB to enter the nucleus and activate transcription of HIV-1 among other targets. If this is the case, we expect that IKK inhibitors would interfere with the activation of latent proviruses. As is also alluded to, that NF-kB upregulates transcription of antiviral responses. Could these be involved in decreasing latency. As this work is written it seems that somehow the investigators came upon IKK inhibitors as a means to modestly enhance expression from latent viruses and they have no idea why. If NF-kB is important for HIV-1 transcription, especially in the initial absence of Tat, the must be stronger factors that counterbalance this effect to start transcription. More discussion is important here.
We agree with the reviewer in that, at first glance, HIV transcription should be blocked upon inhibition of IKK and/or TBK function. However, innate immunity in HIV-1 pathogenesis is best understood in the context of acute infection and recent data suggest that innate immunity can also be used to improve the efficacy of HIV-1 cure strategies (PMID: 34824401). Based on this idea, we explored the role as LRA or LPA of distinct well-described inhibitors of innate immune signalling pathways, including JAKi (PMID: 36131914) and the TBK/IKK inhibitors described in the present work, aimed at report their putative role as latency reactivators. Indeed, although counterintuitive, several evidences from the literature support the idea that the shock phase of the shock-and-kill approach to reverse HIV-1 latency may be achieved in the absence of NF-κB, with the potential to avoid unwanted autoimmune- and or inflammation-related side effects associated with latency-reversing strategies (PMID: 31243131;PMID: 36285453, PMID: 34832672), as reported in our brief report. The description and evaluation of the underlying mechanism of action was out of the scope of the present report. However, following the reviewer’s suggestion we have included a better description of the underlying hypothesis in the introduction and the discussion sections.
Minor points:
- the abbreviation IKKis should be defined.
Definition of IKKis is now included at its first mention (P1; L23).
- VOR was not defined until the figure legend of Figure 1 on page 3 although it's used on page 2.
Corrected (P2; L84).
- line 86 should read "...best-performing IKK inhibitors..." for better clarity.
We have clarified the criteria to choose MRT67307 and TPCA-1 and reformulated the text accordingly (P3; L93-97).
- line 97 "...concentrations where harbour anti-HIV activity..." is missing something (between where and harbour?).
Certainly, there were missing words. The sentence has been corrected to “concentrations where EFV and RAL harbour anti-HIV activity” (P3; L112).
- Figure 1 c D3 is unclear. na?
A description of the meaning of the numbers and “na” has been included in the text of the legend in figure 1 (P4; L147-149).
- in the Discussion section, "modulation" should be defined for clarity. Modulation, of course, can be up- or down-regulation (line 166). NF-kB is mobilized by silencing I-kB alpha. Again, this "activates HIV" (line 168). This is of course the opposite of what would be achieved by inhibiting IKKs... here also (line 170) there is mention of "modulating" TBK. Only IKK inhibitors are used and discussed, so it would be more clear to just say inhibition.
Following the reviewer suggestion, we have clarified the work modulation all over the manuscript, by stating “inhibition” or “activation” when corresponding.
- Line 162: "... immunoblotting: antirabbit..." should read anti-rabbit to be consistent with the other antibody designations.
- Line 272 should read "in the dark".
- line 274 the 5 in 105 should be a superscript.
All typographic errors and grammar and formatting issues have been corrected. Moreover, we have gone through the manuscript to fix all issues regarding, language and reference formatting.